# A tetracycline-dependent ribozyme switch allows conditional induction of gene expression in *Caenorhabditis elegans*

Lena A. Wurmthaler[1,2], Monika Sack[1,2], Karina Gense[2,3], Jörg S. Hartig[1,2] & Martin Gamerdinger[2,3]

The nematode *Caenorhabditis elegans* represents an important research model. Convenient methods for conditional induction of gene expression in this organism are not available. Here we describe tetracycline-dependent ribozymes as versatile RNA-based genetic switches in *C. elegans*. Ribozyme insertion into the 3′-UTR converts any gene of interest into a tetracycline-inducible gene allowing temporal and, by using tissue-selective promoters, spatial control of expression in all developmental stages of the worm. Using the ribozyme switches we established inducible *C. elegans* polyglutamine Huntington's disease models exhibiting ligand-controlled polyQ-huntingtin expression, inclusion body formation, and toxicity. Our approach circumvents the complicated expression of regulatory proteins. Moreover, only little coding space is necessary and natural promoters can be utilized. With these advantages tetracycline-dependent ribozymes significantly expand the genetic toolbox for *C. elegans*.

[1] Department of Chemistry, University of Konstanz, 78457 Konstanz, Germany. [2] Konstanz Research School Chemical Biology (KoRS-CB), University of Konstanz, 78457 Konstanz, Germany. [3] Department of Biology, University of Konstanz, 78457 Konstanz, Germany. Correspondence and requests for materials should be addressed to J.S.H. (email: joerg.hartig@uni-konstanz.de) or to M.G. (email: martin.gamerdinger@uni-konstanz.de)

nducible regulatory systems are very powerful research tools to investigate the cellular function of individual genes. Adapting them to multicellular genetic model organisms is key for understanding gene function during cell differentiation and development, as well as disease progression. Though *C. elegans* is a widely used research animal model, only a few useful methods have been adapted for the temporal and spatial control of gene expression. One method is to use heat-shock-transcription factor-1 (Hsf-1)-driven FLP or Cre recombinases to excise regulatory genetic elements in transgenes to control gene expression[1,2]. However, major drawbacks of this method are the use of heat stress as the inducing trigger that can have an impact on cell physiology and gene function, and the non-reversible switching mode that occurs on the DNA level. Another method employs a transcription-based repressible binary expression system originating from the mold fungus *Neurospora crassa*[3,4]. This system allows inducible and reversible control of gene expression by feeding animals the non-toxic small molecule quinic acid. A disadvantage of this method, however, is the requirement of a fine-tuned and balanced expression of the fungal transcriptional activator-repressor pair in *C. elegans* and the restriction to use a non-authentic promoter driving the gene of interest[3].

Apart from using condition-responsive or effector-responsive transcription factors, small self-cleaving ribozymes have been utilized for controlling gene expression on a post-transcriptional level. For this purpose, ribozymes are inserted into non-coding regions of the mRNA of interest. Cleavage of the respective mRNA results in RNA degradation and hence significant downregulation of gene expression (Fig. 1a). Effector-responsive ribozymes can be engineered by inserting ligand-binding RNA aptamer sequences which control the activity of the RNA catalyst[5–8].

The utilization of ribozyme switches for regulating gene expression in *C. elegans* has several distinct advantages compared to the existing methods. Ribozyme-based approaches are relieved of the necessity of having expressed trans-acting protein factors in a coordinated fashion. The post-transcriptional regulation mechanism is part of the mRNA and hence intrinsically allows for robust expression control. It needs comparably little coding space and hence can be introduced easily into untranslated regions of mRNAs. Further, the gene of interest can be expressed from natural promoters preserving the tissue-specific expression pattern, which is highly relevant if functional studies are carried out.

Here we adapted a tetracycline-dependent ribozyme switch for conditional transgene expression in *C. elegans*. Inserting the ribozyme into the 3'-UTR of genes resulted in robust induction of protein expression upon feeding animals tetracycline in low, non-toxic concentrations. This system allows the temporal control of gene expression during all developmental stages of the animal and is suitable for tissue-selective expression. Using the ribozyme-based switch we established inducible *C. elegans* polyglutamine (polyQ) Huntington's disease models.

## Results

### Ribozymes as tool to control gene expression in *C. elegans*.
Ribozyme-based switches are powerful research tools to achieve conditional gene expression in model organisms like bacteria, *Saccharomyces cerevisiae* or human cell culture[6]. To test whether such systems are adaptable for the animal model *C. elegans*, we utilized a type III hammerhead ribozyme (HHR) from *Schistosoma mansoni*[9] inserted into the 3'-UTR of a pharynx-specific mCherry reporter gene (myo-2p::mCherry). In order to assess the impact of ribozyme cleavage activity on gene expression we inserted a constitutively active and an inactive variant with a

single A-to-G point mutation in the catalytic core of the HHR[9] into the reporter (Fig. 1b). Several independent transgenic strains were constructed carrying the mCherry reporters along with an ubiquitously expressed GFP selection marker (dpy-30p::GFP) (Fig. 1c). In all strains the active HHR efficiently suppressed mCherry expression while strong fluorescence was observed in the pharynx of animals harboring the inactive variant (Fig. 1d). As expected, the mCherry transcript levels were strongly reduced in worms carrying the active HHR-mCherry construct (Fig. 1e), showing that self-cleavage of the HHR leads to mRNA decay and thus decreased protein expression. Thus, this type of ribozyme shows strong activity in *C. elegans* and modulation of its cleavage activity by utilizing ligand-dependent versions could enable conditional gene expression.

### A tetracycline-dependent ribozyme switch in *C. elegans*.
Ribozyme cleavage activity and thus mRNA decay can be regulated by introducing ligand-binding aptamer sequences into the ribozyme, resulting in a so-called aptazyme[10]. Binding of the ligand can either induce (expression OFF-switch) or inhibit (expression ON-switch) ribozyme activity. Because so far no convenient system exists for the induction of gene expression in *C. elegans*, we utilized a tetracycline-regulated aptazyme that was developed by Suess and coworkers as an ON-switch in human cell culture (Fig. 2a)[8]. The aptamer binds its ligand with sub-nanomolar affinity in vitro[11,12] and is thus well suited for usage in an animal model. Moreover, tetracycline is known for a high cell-permeability, and development, stress adaptation, and life span analyses showed that tetracycline is well tolerated by the worms inducing toxicity only at concentrations above 25 μM (Supplementary Fig. 1a–d). The tetracycline-binding aptamer can be linked to the HHR via different RNA linkers, referred to as communication modules, that affect switching performance[8]. We tested four different communication modules that were engineered by Beilstein et al. for conditional gene expression in mammalian cells[8]. The different aptazymes were first tested in *C. elegans* using the myo-2p::mCherry reporter (Supplementary Fig. 2a, b) and the best performing aptazyme was inserted into the 3'-UTR of a ubiquitously expressed mCherry reporter (icd-1p::mCherry). Stable transgenic worms carrying the reporter were generated and treated with low, non-toxic concentrations of tetracycline. Microscopic analysis of worms showed a strong tetracycline-induced increase of mCherry fluorescence throughout the body of the animal (Fig. 2b) and immunoblot analysis confirmed induced mCherry protein levels (Fig. 2c). Confocal microscopy of worms indicated a very broad induction of mCherry fluorescence in multiple tissues, except for the germline, but this was likely due to the general germline-specific chromatin silencing of repetitive multi-copy transgenes as previously observed (Supplementary Fig. 3)[13]. Thus, the aptazyme system works in most if not all tissues of *C. elegans*. A control strain carrying the icd-1p::mCherry reporter with at catalytically inactive aptazyme in the 3'-UTR showed steady, tetracycline-independent mCherry expression (Supplementary Fig. 4a, b). Moreover, tetracycline did not affect endogenous *icd-1* transcript levels (Supplementary Fig. 5). Thus, the inducible mechanism relies on a tetracycline-induced switch in the activity of the RNA catalyst. To get a more precise quantification of gene induction by tetracycline we conducted worm flow cytometry analyses. Synchronized L1 larvae were fed tetracycline in different concentrations and mCherry fluorescence was recorded every 24 h until adulthood. Protein expression reached a plateau for concentrations above 5 μM of tetracycline and maximal induction (2.5-fold ± 0.26 s.d.; $n > 300$) occurred after 48 h (Fig. 2d, Supplementary Fig. 6). RT-qPCR analyses confirmed a rapid tetracycline-induced response on the mRNA

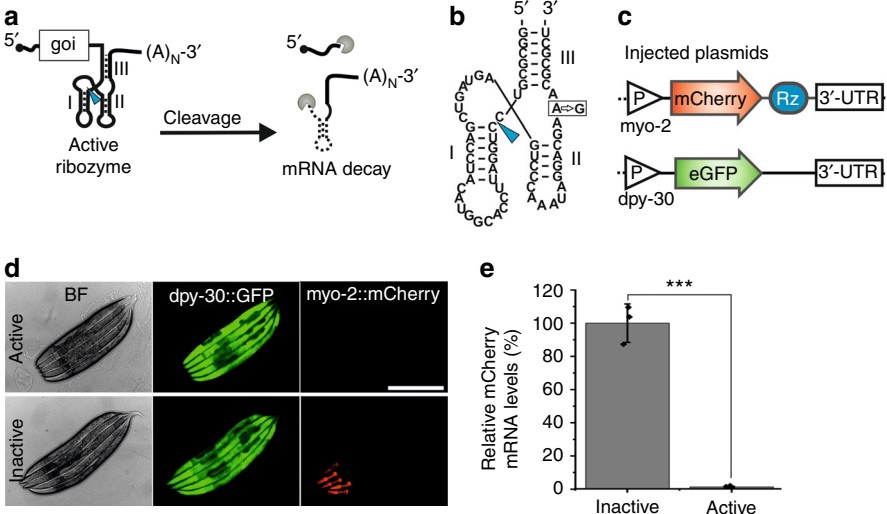

**Fig. 1** Impact of self-cleaving ribozymes on gene expression in *C. elegans*. **a** Schematic showing how ribozymes in the 3′-UTR affect mRNA stability. Blue arrow indicates self-cleavage site. (A)$_N$ indicates poly-A tail. goi, gene of interest. **b** Sequence and secondary structure of type 3 hammerhead ribozyme with characteristic stems I–III. Blue arrow indicates cleavage site. Box indicates inactivating A-to-G mutation. **c** Schematic showing constructs injected into worms. Active and inactive ribozyme variants were inserted into the myo-2p::mCherry construct. P promoter. Rz ribozyme. **d** Microscope images of young adult worms injected with plasmids shown in **c**. BF, bright field. Scale bar, 300 μm. **e** RT-qPCR analysis showing the relative mCherry mRNA levels in young adult animals as shown in **d**. GFP was used as a reference gene. Error bars, s.d. ***p < 0.001 (two-tailed t-test); n = 3. Source data are provided as a Source Data file

level, showing significantly up-regulated mCherry transcript levels (2.2-fold ± 0.54; n = 3) already after 12 h treatment with 10 μM tetracycline, and transcript levels increased further with maximal induction (3.75-fold ± 0.85; n = 3) after 48 h (Supplementary Fig. 7). Moreover, worm flow cytometry analysis revealed that the aptazyme-based system is suitable to induce gene expression in all developmental stages of the worm (L1–L4/YA), as well as in fully developed adults by a 24 h tetracycline treatment (10 μM) (Fig. 2e, Supplementary Fig. 8). These data show that a robust inducible gene expression system can be generated in *C. elegans* with a single genetic modification by inserting the 132 nucleotide aptazyme sequence into the 3′-UTR of a gene of interest.

**Aptazyme-controlled tissue-selective gene expression.** Next, we tested the functionality of our ribozyme-based switch to induce tissue-selective gene expression. *C. elegans* strains were constructed carrying aptazyme-regulated mCherry reporters driven by tissue-specific promoters for the pharynx (myo-2p), body wall muscles (myo-3p) and neurons (rab-3p). In all strains tetracycline treatment specifically induced reporter gene expression in the expected tissue cells (Fig. 3a, b). Interestingly, the mCherry expression was induced uniformly in the large body wall muscle tissue from head to tail. This indicates that tetracycline induces an even inhibition of ribozyme activity throughout the anterior-posterior axis of the worm. Moreover, as expected the pan-neuronal targeted reporter showed strongest mCherry fluorescence in the anterior nerve ring, which constitutes the largest assembly of neurons in *C. elegans*. In addition, mCherry expression was also induced throughout the other regions of the nervous system including the ventral and dorsal nerve cord and tail ganglion (Fig. 3a). Control experiments showed no effect of tetracycline on the endogenous transcript levels of *myo-2*, *myo-3*, and *rab-3*, confirming that tetracycline induced the tissue-selective reporter gene expression only by acting on the inserted aptazyme (Supplementary Fig. 8). These data show that the aptazyme-based system is applicable for spatial control of transgene expression in various tissues.

**Aptazyme-inducible Huntington's disease models.** Inducible expression systems allow for generating animal models of human disease that rely on temporal expression of highly toxic components. Thus, to test whether the ribozyme switch can be applied to generate inducible *C. elegans* disease models, we inserted the aptazyme sequence into the 3′-UTR of constructs expressing human Huntingtin exon 1 (Htt) containing an extended poly-glutamine (polyQ) tract that causes Huntington's disease in humans. A polyQ stretch above a critical threshold (~35Q) renders the protein aggregation-prone and toxic, and the length of the polyQ expansion inversely correlates with the age of disease onset in humans[14,15]. We generated *C. elegans* strains expressing an aggressive Htt variant fused to mCherry harboring a pathogenic stretch of 109Q. Control strains were constructed expressing Htt with a polyQ repeat below the pathogenic threshold (25Q). We generated two different disease models exhibiting either ubiquitous (icd-1p) or neuron-specific (rab-3p) Htt expression (Fig. 4a, b). Tetracycline treatment increased mCherry fluorescence in all strains, however, as expected only in the Htt109Q strains the mCherry signal was found concentrated in large inclusion bodies (Fig. 4c-f). In the ubiquitous model Htt109Q::mCherry aggregates formed in various tissues including the pharynx, intestine, muscle and nervous system (Supplementary Fig. 9a, b). In the pan-neuronal Htt model the inclusion bodies appeared as expected only in the neuronal tissue including the chemosensory neurons, anterior nerve ring, ventral nerve cord and tail ganglion (Supplementary Fig. 9c, d). Huntington's disease is a neurodegenerative disorder characterized by a severe lack of motor coordination in humans[15]. Several constitutive muscle-specific or neuron-specific polyQ disease models in *C. elegans* show a strong motility defect[16–19]. Thus, we tested whether the expression of Htt109Q in worms in our inducible models resulted in similar locomotion defects. Indeed, in the ubiquitous Htt109Q strain tetracycline treatment induced a premature paralysis phenotype, whereas Htt25Q and wildtype N2 worms were unaffected (Fig. 4g). Furthermore, to test whether our inducible Htt worm models also recapitulate the neurotoxic aspects of Huntington's disease we examined the thrashing rate of

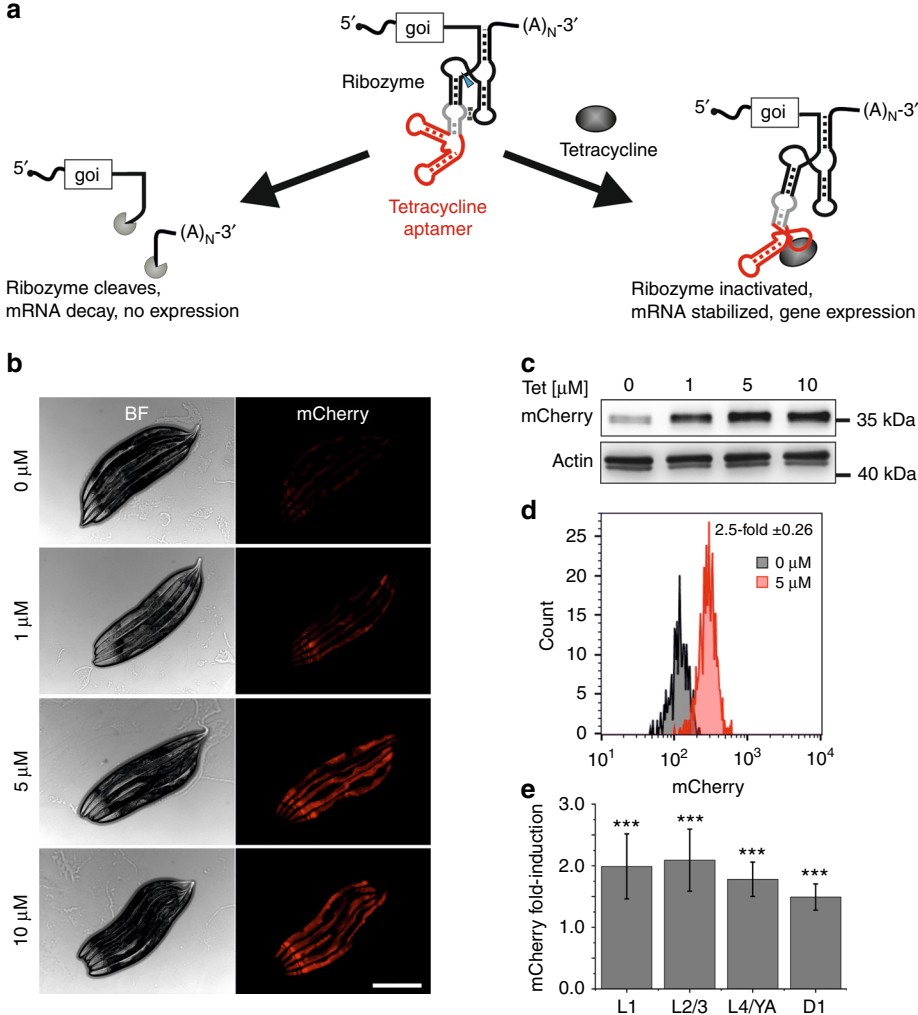

**Fig. 2** Tetracycline-dependent ribozymes enable conditional gene expression in *C. elegans*. **a** Schematic of the tetracycline-induced ribozyme expression ON-switch. A communication module (gray) connects the ribozyme to a tetracycline-binding aptamer (red). Self-cleavage (blue arrow) in the absence of the ligand leads to mRNA decay. Binding of tetracycline induces a conformational change in the ribozyme and its inactivation leading to mRNA stabilization and gene expression. (A)$_N$ indicates poly-A tail. goi, gene of interest. **b** Microscope images of stable transgenic worms carrying aptazyme-regulated icd-1p:: mCherry reporter. Worms were treated with indicated tetracycline concentrations from hatch until adulthood (3 days, 20 °C). BF, bright field. Scale bar, 300 μm. **c** Immunoblot analysis of mCherry protein levels in animals as shown in **b**. Actin served as loading control. Tet, tetracycline. **d** Flow cytometry analysis of worms carrying aptazyme-regulated icd-1p::mCherry reporter. Synchronized L1 larvae were treated with 5 μM tetracycline for 48 h at 20 °C. Histogram plots mCherry fluorescence intensity against the number of worms. n > 300. **e** Worm flow cytometry analysis as in **d**. Animals were treated at indicated developmental stages with 10 μM tetracycline for 24 h at 20 °C. Bar diagram shows the fold-induction of mCherry fluorescence compared to control. Error bars, s.d. ***$p < 0.001$ (two-tailed *t*-test); n > 300. L1-4 larval stage 1-4. YA young adult. D1 day 1 adult. Source data are provided as a Source Data file

rab-3p::Htt109Q/25Q::mCherry animals swimming in a drop of buffer. This assay is in particular useful for measuring the dysfunction of motor neurons that constitute about a third of all *C. elegans* neurons[16]. Tetracycline treatment significantly reduced the number of body bends in Htt109Q worms while no effect was observed in the Htt25Q strain (Fig. 4h). Thus, the induced expression of Htt109Q exerted neurotoxic effects leading to impaired body movement. In sum these data show that the aptazyme-based conditional gene expression system is applicable to generate inducible *C. elegans* disease models that underlie expression of highly toxic proteins.

**Aptazyme-controlled rescue of mutant phenotypes.** Finally, we tested the robustness of our aptazyme system in a physiological context by rescuing a mutant phenotype. Therefore we used the *unc-119(ed3)* mutant strain which shows a severe uncoordinated

(unc) locomotion phenotype due to a developmental defect of the nervous system[20,21]. Transgenic strains in the *unc-119(ed3)* background were constructed carrying extrachromosomal arrays either with a constitutive Cbr-unc-119 rescue plasmid or an inducible version thereof harboring the tetracycline-dependent aptazyme in the unc-119 3'-UTR (Fig. 5a). As expected, the mutant phenotype was rescued in all transgenic animals harboring the constitutive plasmid, whereas most worms carrying the aptazyme-regulated rescue plasmid still showed the unc-119 mutant phenotype in the absence of tetracycline. Tetracycline treatment (10 μM) significantly rescued the unc phenotype in the strain carrying the inducible Cbr-unc-119 plasmid, while no effect was observed in the parental *unc-119(ed3)* strain (Fig. 5b). Thus, the aptazyme system is suitable to control complementation experiments of mutant alleles, enabling a precise analysis of gene function during *C. elegans* development.

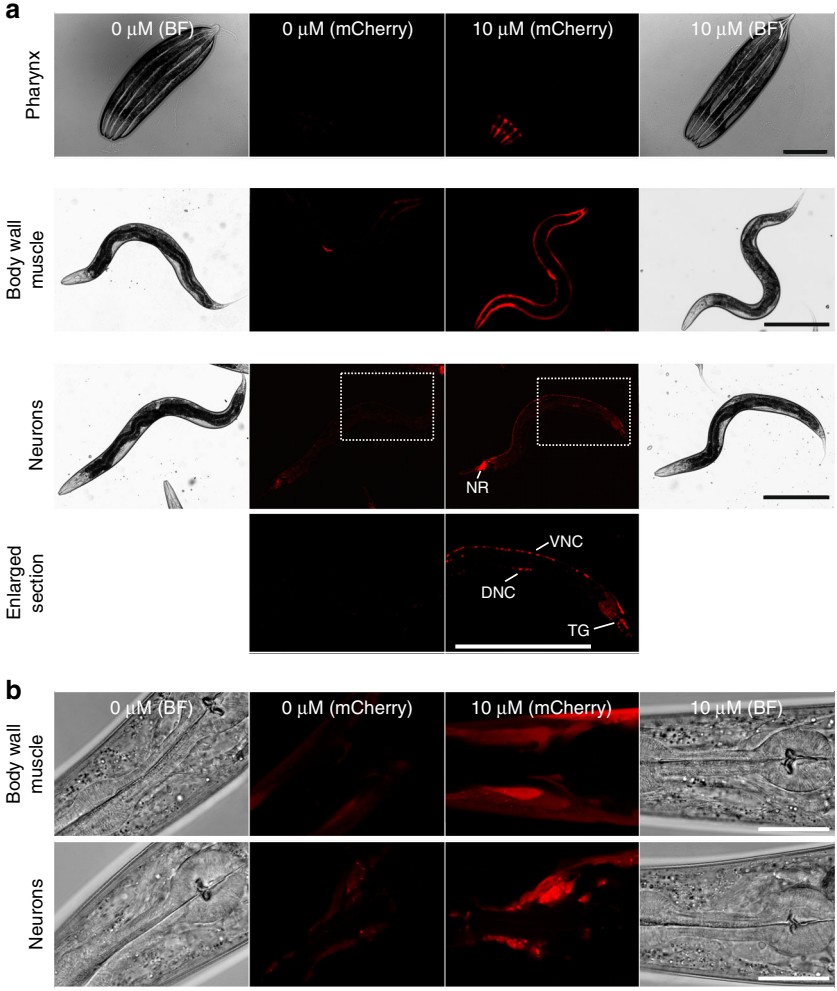

**Fig. 3** Aptazymes allow tissue-selective induction of gene expression in *C. elegans*. **a** Microscope images of worms carrying 3′-UTR aptazyme-regulated mCherry reporters driven by tissue-specific promoters for the pharynx (myo-2p), body wall muscles (myo-3p) and neurons (rab-3p). Worms were treated with 10 μM tetracycline from hatch until adulthood (3 days at 20 °C). Boxed sections in images of worms carrying the pan-neuronal reporter are shown enlarged with adjusted contrast to visualize the ventral nerve cord (VNC), dorsal nerve cord (DNC) and tail ganglion (TG). NR nerve ring, BF bright field. Scale bar, 300 μm. **b** Confocal microscope images of animals as in **a**. Images show the head region of worms carrying aptazyme-regulated mCherry reporters expressed in body wall muscles and neurons. BF bright field. Scale bar, 30 μm

## Discussion

The major advantage of the presented aptazyme method to control transgene expression in *C. elegans* is its simplicity. A single small genetic modification of the 3′-UTR converts any gene of interest into an inducible gene allowing temporal and spatial control of expression in all developmental stages of the worm. Thus, this technique has great advantages over existing approaches which need to integrate multiple components into the *C. elegans* genome[1–3]. Moreover, the lack of foreign trans-acting factors (transcriptional silencer and activator) allows gene regulation without concern for their potential toxicity, which makes this technology also very attractive for human gene therapy due to lack of adaptive immune responses.

The reported ribozyme switch provides a conditional gene expression platform that works in a physiologically relevant dynamic range sufficient to address important biological questions, as exemplified by the inducible Huntington's disease models and the *unc-119(ed3)* rescue experiments. The aptazyme switching performance in *C. elegans* is comparable to those observed in other eukaryotic model systems like human cells[8].

Nevertheless, the dynamic range with maximal 3.75-fold ± 0.85 fold gene induction (Supplementary Fig. 7) is weaker compared to the classical use of heat-shock promoters[22]. The main advantage of our system over the use of heat shock promoters, however, is that natural promoters can be used to drive gene expression and that gene function can be precisely analyzed under non-stress conditions. Nevertheless, depending on the biological question asked a higher switching efficiency might be necessary in *C. elegans*. Integrating additional ligand-sensing domains or inserting aptazymes in tandem may improve switching performance. Besides the 3′-UTR, aptazymes are also expected to be effective when inserted into the 5′-UTR and introns of mRNAs. In addition to tetracycline several other ligands like theophylline, guanine or thiamin pyrophosphate (TPP) have been utilized to regulate HHR cleavage activity[23–26]. The corresponding aptamers bind these ligands with low nanomolar affinity[27–29] similar to the utilized tetracycline aptamer and could therefore be functional in *C. elegans* as well. Implementing different ligand-dependent ribozyme switches could enable independent temporal control of two or more genes of interest in the same animal. Our study lays

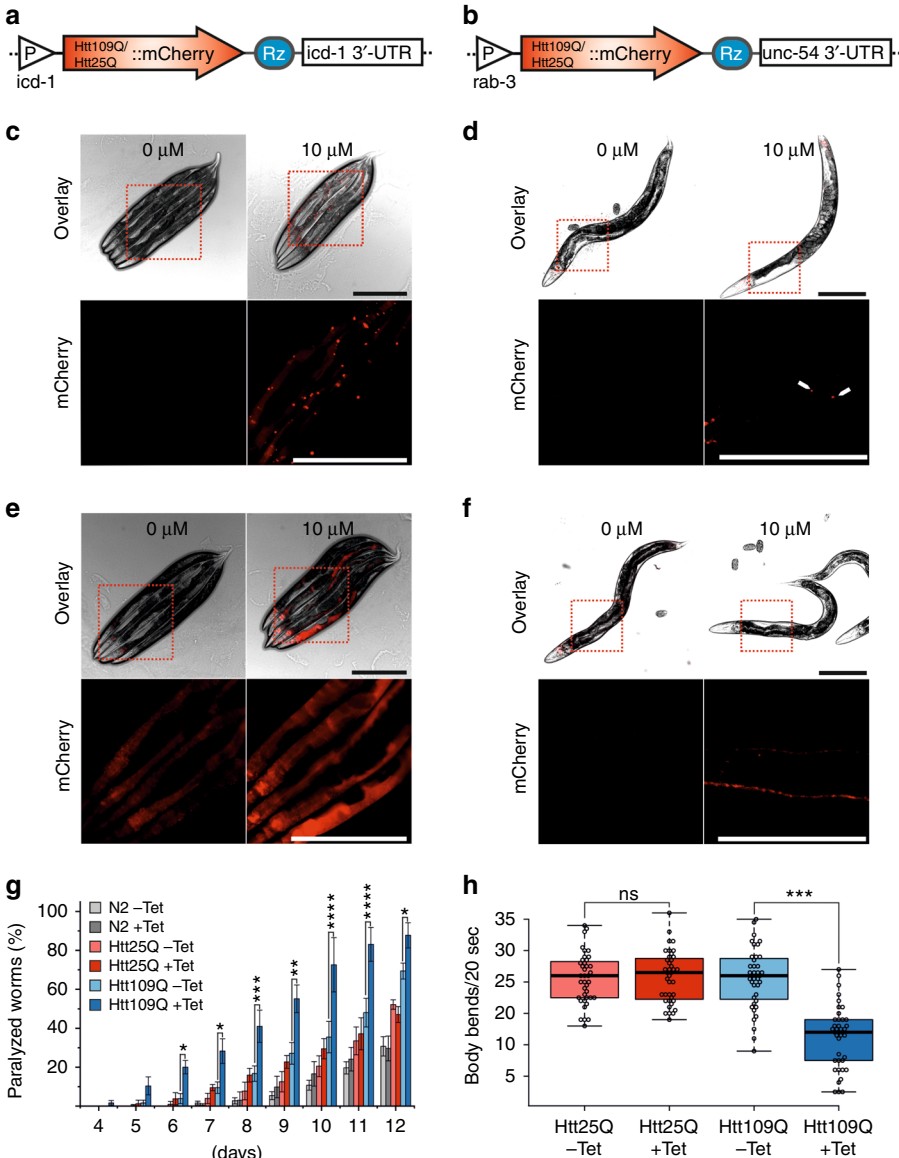

**Fig. 4** Aptazyme-inducible proteotoxic *C. elegans* disease models. **a**, **b** Schematics showing constructs for inducible expression of human Huntingtin exon 1 (Htt) containing a polyglutamine stretch of either 109 (Htt109Q) or 25 glutamines (Htt25Q) fused to mCherry driven either by the ubiquitous promoter icd-1p (**a**) or the pan-neuronal promoter rab-3p (**b**). Rz, ribozyme. **c** Microscope images of worms carrying icd-1p::Htt109Q::mCherry construct. Animals were fed 10 μM tetracycline from hatch until adulthood (3 days, 20 °C). Upper row shows overlay of bright field images with mCherry fluorescence. Fluorescence signal in boxed sections is shown enlarged to visualize Htt109Q::mCherry aggregates (bottom row). Scale bar, 300 μm. **d** Same analysis as in **c** but with rab-3p::Htt109Q::mCherry strain. White arrows indicate Htt109Q::mCherry inclusion bodies. **e**, **b** Same analysis as in **b** and **c** but with Htt25Q::mCherry worms, respectively. **g** Strains carrying the constructs shown in **a** were treated with 10 μM tetracycline from hatch. N2 worms served as wildtype control. Diagram shows the percentage of paralyzed worms at indicated days of adulthood. Error bars, s.d. *$p < 0.05$, **$p < 0.01$, ***$p < 0.001$, ****$p < 0.0001$. Two-Way ANOVA, Bonferroni post-hoc test; $n = 3$. **h** Strains carrying the constructs shown in **b** were treated with 10 μM tetracycline from hatch until day 2 of adulthood (4 days at RT). Box plot shows the number of body bends made by worms in a drop of liquid in 20 s. ***$p < 0.001$ (two-tailed *t*-test); ns not significant; $n = 40$. Center line = median; box length = upper + lower quartile; whiskers = minimum/maximum quartile. Source data are provided as a Source Data file

the groundwork for future studies aiming to adapt ribozyme switches to control gene expression in this important research animal model.

## Methods

**Ribozyme constructs**. Active and inactive variants of the type III HHR from *Schistosoma mansoni*[9] were inserted into plasmid pCFJ90 (myo-2p::mCherry)[30] by standard cloning techniques into the 3'-UTR 16 nt downstream of the stop codon. In the same manner, the tetracycline-dependent aptazyme sequences carrying different communication modules[8] were placed into the 3'-UTR of plasmid pCFJ90. In all constructs, aptazymes were insulated by 5'-CAAACAAACAAA and

3'-CAAACAAACAAA spacers (Supplementary Note 1). The aptazyme sequence containing the communication module K4[8] was additionally inserted into the 3'-UTRs of plasmids pCFJ104 (myo-3p::mCherry), pGH8 (rab-3p::mCherry)[30] and pIK137 (unc-119p::Cbr-unc-119)[31]. The ubiquitously expressed mCherry reporter (icd-1p::mCherry) was constructed by exchanging the promoter and 3'-UTR of pCFJ90 with those of the *icd-1* gene amplified by PCR from genomic DNA. The K4 aptazyme sequence was placed into the *icd-1* 3'-UTR 16 nt downstream of the stop codon. The inducible polyQ-Htt plasmids were constructed by subcloning Huntingtin exon 1 encoding sequences from plasmids p109QHtt.EGFP-N1 and p25QHtt.EGFP-N1[32] into the icd-1p::mCherry and rab-3p::mCherry vectors. Details about ribozyme sequences and constructs are available in Supplementary Note 1. All plasmids are available upon request.

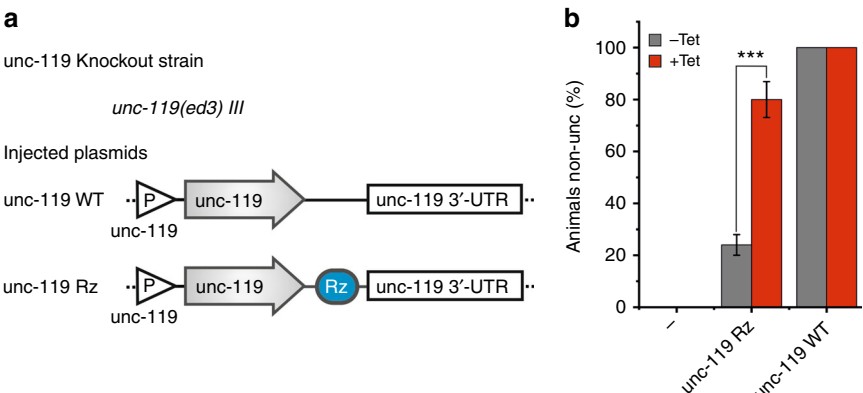

**Fig. 5** Aptazymes enable rescue control of mutant phenotypes in *C. elegans*. **a** Schematics showing constructs injected into *unc-119(ed3)* strain. A constitutive unc-119p::Cbr-unc-119 rescue plasmid (unc-119::WT) and a tetracycline-inducible unc-119p::Cbr-unc-119 rescue construct with the aptazyme in the 3′-UTR (unc-119::Rz) were used to rescue the unc phenotype of *unc-119(ed3)* worms. Rz ribozyme. **b** Parental *unc-119(ed3)* worms (−) and transgenic strains carrying the constructs shown in **a** were treated with 10 μM tetracycline from hatch. Diagram shows the percentage of unc worms at day 1 of adulthood. Twenty-four animals per group were used and experiments were carried out in biological triplicates. Error bars, s.d. ***$p < 0.001$ (two-tailed *t*-test); $n = 3$. Source data are provided as a Source Data file

**Strains and transformation**. *C. elegans* Bristol strain N2 and EG6701 (*unc-119 (ed3)III*) were used for transformations. Worms were cultured according to standard techniques with *E. coli* OP50 as food source[33]. Transgenic strains were generated using standard microinjection protocols[34]. For each transformation, at least two independent transgenic lines carrying extrachromosomal arrays were obtained showing similar results. In all transformations pPD152.79 (dpy-30p::GFP; gift from Andrew Fire, Addgene plasmid #1704) was used as a co-injection marker. Stable transgenic lines were generated by X-ray mediated integration of extrachromosomal arrays (20 gy; X-RAD 225iX) followed by at least 2x backcrossing to N2 worms. Strain information is available in Supplementary Note 2. The study complies with all relevant ethical regulations for animal testing and research.

**Tetracycline treatment**. Tetracycline (Sigma-Aldrich, 87128) was dissolved in sterile water (500 μM stock solution) and added to standard nematode growth medium (NGM) immediately prior to pouring plates (50–55 °C). Alternatively, tetracycline was added to solid NGM plates and plates were allowed to dry over night in the dark at RT. Plates were seeded with a 20x concentrated *E. coli* OP50 culture and stored in the dark at 4 °C and used within one week. Worms carrying aptazyme-regulated reporters were grown on tetracycline plates from hatch for three days at 20 °C before assessment of mCherry expression. Therefor a timed egg-lay on tetracycline plates was performed for 6 h. For flow cytometry analyses worms were synchronized by hypochlorite bleaching and cultivated at 20 °C in liquid culture and harvested by 30% sucrose cushion floatation in 0.1 M NaCl as previously described[35]. Tetracycline was added to liquid cultures at indicated developmental stages directly from a 1000x stock solution made in ethanol.

**Microscopy**. Worms were immobilized with 1% sodium azide on a 3% agarose pad and directly imaged using a DM6000B-Cs microscope (Leica) equipped with a DFC 365FX camera (Leica). For analysis of tissue-selective mCherry expression worms were anesthetized with 25 mM levamisole (LKT Laboratories) and fluorescence was assessed with a confocal laser-scanning microscope TCS SP8 (Leica) with a 63x objective. Images were adjusted as necessary in Fiji (ImageJ) using cropping, brightness and contrast tools.

**Worm flow cytometry**. Worm populations were cultivated in liquid culture and analyzed using a COPAS FlowPilot (Union Biometrica). Data were processed using FlowJo10 software (provided by the FlowKon facility, University of Konstanz). Fluorescence of mCherry was recorded in worms that were gated using time of flight (TOF) and GFP fluorescence as parameters (Supplementary Fig. 6a). Age-matched wildtype N2 worms were used to subtract auto-fluorescence background before calculation of the fold-induction.

**Paralysis and thrashing assays**. Paralysis in each strain (50–70 worms) was tested on plates with or without 10 μM tetracycline from hatch. In the young adult stage worms were placed on plates containing 150 μM 5-fluorodeoxyuridine to inhibit progeny. Screening of paralyzed worm started at day 1 of adulthood. Scored as paralyzed were worms that failed to undergo a full body wave propagation upon repeated prodding with a platinum wire worm picker. Non-paralyzed worms were transferred to fresh plates on day 6. The experiment was repeated twice. For the thrashing assay worms were treated with 10 μM tetracycline from hatch until day 2 of adulthood (4 days at RT). Single worms were transferred to a 96 well round

bottom plate containing 10 μl M9 buffer (85.6 mM NaCl, 4.2 mM $Na_2HPO_4$, 2.2 mM $KH_2PO_4$, 1 mM $MgSO_4$) and immediately videotaped for 20 s using a MZ10F microscope (Leica) equipped with a DFC 3000 G camera (Leica). Quantification of body bends was performed with Fiji (ImageJ) using the wrMTrck plugin[36].

**Life span analysis**. Life span experiment was performed with N2 wild type worms on OP50 seeded plates with or without 10 μM tetracycline from hatch. In the young adult stage 100 worms were placed on plates containing 150 μM 5-fluorodeoxyuridine to inhibit progeny. Screening of live/dead worms started at day 4 of adulthood. Live worms were transferred to fresh plates on day 10 of adulthood.

**Thermotolerance assay**. N2 wild type worms were cultivated on plates with or without 10 μM tetracycline from hatch at 20 °C. After 2 days 50 L4 worms were placed on fresh plates, and the following day plates were sealed with parafilm and incubated at 34 °C for 2–7 h in a water bath. Live/dead worms were scored after 16–17 h recovery at 20 °C.

**Unc-119 rescue experiment**. Transgenic strains in the *unc-119(ed3)* background were cultivated on plates with or without 10 μM tetracycline from hatch at 20 °C for 3 days. Single worms were transferred to a 96 well round bottom plate containing 10 μl M9 buffer and immediately videotaped for 20 s using a MZ10F microscope (Leica) equipped with a DFC 3000 G camera (Leica). Worms carrying out less than 10 body bends were considered as *unc*. 24 animals per group were used and experiments were carried out in biological triplicates.

**Immunoblot analysis**. Total lysates were prepared by sonication of worms in SDS lysis buffer (62.5 mM Tris-pH 6.8, 1 mM EGTA, 2% SDS, 10% sucrose) supplemented with protease inhibitor cocktail (Roche). Samples were boiled for 5 min and applied to SDS-PAGE and electroblotted onto a nitrocellulose membrane as previously described[35]. Blots were probed with monoclonal anti-mCherry antibody (1:1000; Novus Biological, NBP1-96752) and monoclonal anti-Actin (1:5000; Santa Cruz, sc-47778).

**RT-qPCR analysis**. Lysates were prepared by sonication of worms in RLT buffer (Qiagen) and total RNA was extracted using the RNAeasy Mini Kit (Qiagen) according to manufacturer's protocol. RNA was reverse transcribed using the QuantiTect Reverse Transcription kit (Qiagen) in a total volume of 10 μL. GoTaq qPCR Master Mix (Promega) was used for qPCR in a total volume of 10 μL (1 μL cDNA, 25 μM primer). qPCR was performed on a Biorad CFX cycler using a two-step protocol (15 s at 95 °C/60 s at 60 °C) for 40 cycles following a 2 min hot-start activation at 95 °C. PCR specificity was confirmed by melting curve analysis (65–95 °C) and data was analyzed using the BioRad CFX software. Data were analyzed using the comparative $2^{ΔΔCt}$ method and *gfp* or *ama-1* as reference genes, as indicated. QPCR primers are listed in Supplementary Note 3.

**Reporting Summary**. Further information on experimental design is available in the Nature Research Reporting Summary linked to this Article.

## Data availability
The source data underlying this study are provided as a Source Data file or from the authors upon reasonable request.

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

## Acknowledgements
This work was supported by the German Research Foundation (DFG: CRC 969 "Chemical and Biological Principles of Cellular Proteostasis", TP A5 to J.S.H., TP A7 to M.G.). We thank Renate Schlömer for technical assistance.

## Author contributions
M.G. and J.S.H. conceived the study. M.G., J.S.H., and L.A.W. designed the studies and wrote the manuscript . L.A.W., M.S., and K.G. performed experiments and analyzed data.

## Additional information

**Competing interests:** The authors declare no competing interests.

