## [Peer Review File · Nature Communications]

Reviewers' Comments:

Reviewer #1:

Remarks to the Author:

This manuscript by Wurmthaler et al. and Gamerdinger is a generally well-written manuscript that presents very strong support for the function of engineered hammerhead ribozyme switches as gene control devices in *C. elegans*. This is an important technology demonstration that appears to be sufficiently functional for practical applications. Indeed, the authors nicely demonstrate precise regulation of gene expression in response to the addition of tetracycline. The experiments appear to be well designed and the results appear to be solid. Although the 'aptazyme' concepts and constructs are not novel, it is still good to see such a strong demonstration of the utility of this emerging technology in a model eukaryote. I agree with the authors that this technology will have much utility as a research tool, and I fully support publication in Nature Communications. I only have a few very minor comments, which the authors and the editors can largely choose to ignore without any substantive drawbacks.

1. In the Abstract, the authors comment on "temporal and spatial" control, and I first believed that the ribozyme was responsible for both. However, the ribozyme only permits temporal control, whereas spatial expression presumably is due to the promoter. Perhaps the authors could be clearer about this issue.
2. Line 62, the phrase "only few" would be better written as "only a few".
3. Throughout the manuscript, "type 3" should be written as "type III". Likewise, the stems should be named I, II and III.
4. Line 135, I think it is not correct to state that "the integrated aptamer binds its ligand with sub-nanomolar affinity". The published binding data appear to be for the aptamer alone. Once integrated with the hammerhead, I suspect the affinity becomes much poorer.
5. Line 137, the phrase "and well" should be written as "and is well".
6. The term "communication module" should be defined on first use.
7. Line 212, "trashing" should be "thrashing".
8. On line 230, I agree that simplicity is the major advantage for worms, but the lack of an adaptive immune response would be far more important for organisms such as humans. Perhaps the authors might add a sentence to note this important problem of protein-based gene regulation systems.
9. My greatest concern about the current manuscript is that it took me a lot of time to determine that the authors had not designed the tetracycline-dependent ribozymes, but rather they used sequences that are identical to those published in reference 8. The authors do cite this previous work in the appropriate places, but I think they should add a sentence or two that makes it very clear where these sequences were first published.
10. Related to Point #9, the authors should have a better graphic for the constructs they used. The RNA constructs are the molecules that other researchers will want to use, but they will have a hard time reconstructing the proper sequence from the fragments presented in Supplementary Figure 1C, or even from the sequences presented in Supplementary Note 1. For example, it is not readily evident from Supplementary Figure 1C what the orientation of the communication modules should be. With a better graphic, readers who wish to use the RNA switches will be much happier.
11. Many of the references have inaccurate formatting of journal names.

Reviewer #2:

Remarks to the Author:

In this manuscript Wurmthaler and colleagues, report the development of an RNA-based genetic system for conditional gene expression in *C. elegans*. This system is a ribozyme switch capable of conferring tetracycline-dependent gene expression upon insertion into the 3'-UTR of the gene of interest. Specifically, the tetracycline-dependent ribozyme switch is constructed by introducing a tetracycline-dependent aptazyme sequence (i.e. tetracycline-binding aptamer sequences) into the 3'-UTR of the gene of interest. Importantly, gene expression can be regulated by tetracycline

below its toxicity causing concentration. The system can be used to control tissue specific expression as well as for the study of disease models. The authors tested the efficiency of their system by using the *myo-2::mCherry* reporter and inserted the best performing aptazyme into the 3'-UTR of a ubiquitously expressed mCherry reporter (*icd-1::mCherry*). Transgenic animals expressing the reporter showed an increase mCherry fluorescence following tetracycline treatment. Furthermore, by using tissue-specific promoters, they found that their ribozyme-based system allows tissue specific induction of gene expression. Moreover, the aptazyme-inducible system was used to generate a conditional aggressive nematode model of Huntington's disease by expressing an aggressive Htt109Q pathogenic variant fused to mCherry either ubiquitously (*icd-1* promoter) or specifically in neurons (*rab-3* promoter). Overall, this approach expands the genetic toolbox in *C. elegans*. However, the method is not novel since it is already established and used previously in other systems (Beilstein et al., 2015 ACS synthetic biology; Felletti & Hartig, 2017 RNA).

Comments

Ribozyme-based switches enabling the conditional modulation of gene expression are rarely reported in *C. elegans*. Interestingly, an aptazyme as an artificial riboswitch in the nematode has been described in the PhD thesis of Benedikt Klausner (RNA Synthetic Biology using the Hammerhead Ribozyme: Engineering of Artificial Genetic Switches, 2015). Nevertheless, the utilization of such systems has been extensively reported in bacteria, yeast and mammalian cells, therefore compromising the conceptual novelty of the strategy presented in this study (Auslander et al, Mol Biosystems, 2010; Wieland et al, Methods, 2012; Felletti et al, 2016, NatCommun, 10.1038/ncomms12834, e.t.c.)

Besides its implementation for ubiquitous and tissue-specific gene expression, the tetracycline-dependent ribozyme switch was validated by generating a nematode model of Huntington's disease. A point of consideration is the lack of statistically significant difference between the number of paralyzed Htt109Q- expressing animals treated with tetracycline and the non-treated controls (Figure 4g). Please, include a statistical analysis. Statistics are missing from the following figures: Figure 4g, Supplementary Figure 1b and 3b-d.

To strengthen their claims regarding the robustness of this aptazyme-based genetic tool, the authors should test whether it can efficiently control the temporal expression of genes known to be required for proper embryonic function or neuronal development. In other words, they should test the robustness of their system in a physiological context.

The authors should broaden their analyses regarding the side effects of tetracycline in various contexts including ageing, stress conditions, behavior etc. Tetracycline is a strong antibiotic and given the importance of intestinal microbes on health and physiology, it is necessary to validate its effects on worm' physiology.

The authors show that HHR (hammerhead ribozyme) insertion into 3' UTR of *myo-2::mcherry* is sufficient for mcherry suppression. However, all constructs/strains used by the authors integrate HHR only into the 3'UTR of *unc-54*. To further evaluate the efficiency of their system, the authors should use translational reporters inserting the HHR into the 3' UTR of genes of interest. Towards this direction, it would be interesting to evaluate the efficiency of their system by targeting GFP instead of mcherry.

In fig 1d the authors compared transgenic animals injected with *myo-2::mcherry* active and inactive ribozyme variants. However, the authors do not provide evidence regarding the mRNA levels of mcherry between the two transgenic animals carrying either active or inactive ribozyme variants. Even integrated extrachromosomal arrays of two variants may result in different mRNA levels due to random integration into the genome. Therefore, the authors should perform RT PCR showing the mRNA levels of mcherry between these transgenic animals prior to comparing fluorescence levels of mcherry.

In fig 2b the authors show tetracycline-induced expression of mcherry in transgenic animals injected only with the active ribozyme variants. To further verify their observations, monitoring tetracycline-induced expression of mcherry in both active and inactive ribozyme variants is required. Towards this direction, transgenic animals expressing only myo-2::mcherry should be used as controls in the absence and presence of tetracycline to exclude any intervention of tetracycline on myo-2 transcriptional levels per se. Within this scope, mRNA levels of myo-2 upon tetracycline supplementation should be monitored.

In fig 3a,b authors show tetracycline-induced mcherry expression in neurons, pharynx and muscles. Again, the effect of tetracycline supplementation in both active and inactive ribozyme variants is missing. Furthermore, authors should check whether tetracycline supplementation affects or not the transcript levels of the promoters myo-3 and rab-3 tested.

Tetracycline causes toxicity in *C. elegans* animals but authors use concentrations that are below the compound toxicity threshold. On the other hand, utilization of this system chronically (i.e. for lifespan experiments) may cause high toxicity due to accumulation of cellular damage. Have authors tested whether this system affects the lifespan and healthspan of wild type animals?

The authors report that induction of expression is initiated 24h post tetracycline treatment, reaching a peak 48h post treatment. This has three main drawbacks: 1. Expression is not "on" and "off" but is intermediate at least for 48hours until it reaches it's a peak. This means that the conditions of expression are not well controlled and the researcher is not able to know the exact expression level of the gene of interest. The authors have not tested whether this 48hour-period needed to get the peak in protein expression is dependent on the genetic background and the protein/mRNA of interest and 3. 48h is a long period for a drug-inducible system. It would be interesting if by initially providing much higher drug concentrations, they could rapidly induce gene expression and then by using much milder ones simply maintain expression.

The expression of the co-transformation marker is dramatically altered in Figure 1d. This is not expected and may be evidence for multiple secondary effects of this system.

Minor points

Please, provide better quality images to show the body wall muscle- and neuron- specific expression of 3'-UTR aptazyme-regulated mCherry (Figure 3).

In fig 4c-f control conditions are missing.

In Figure 2, panels b and c do not perfectly match to each other. For example, in panel b mCherry is more than doubled in 5 μ M compared to 1 μ M treatment, but in panel c this is not evident.

Figure 4: Why did the authors use different 3'UTRs for *icd-1* and *rab-3* promoter?

Figure 4d and f: Higher magnification images are required for neuronal expression of *htt109Q* and *Htt25Q*, to better visualize diffuse expression/aggregates.

Figure 4g: Mobility defects must be tested also in the lines with the neuronal expression of *Htt109Q* and *Htt25Q*.

Please provide information and reference on previous Huntington's Disease model strains in *C. elegans* (constructs and tissue of expression, as well as motility defects in each case).

Please, use proper *C. elegans* nomenclature.

Response to reviewers' comments

Reviewer #1 (Remarks to the Author):

This manuscript by Wurmthaler et al. and Gamerdinger is a generally well-written manuscript that presents very strong support for the function of engineered hammerhead ribozyme switches as gene control devices in *C. elegans*. This is an important technology demonstration that appears to be sufficiently functional for practical applications. Indeed, the authors nicely demonstrate precise regulation of gene expression in response to the addition of tetracycline. The experiments appear to be well designed and the results appear to be solid. Although the 'aptazyme' concepts and constructs are not novel, it is still good to see such a strong demonstration of the utility of this emerging technology in a model eukaryote. I agree with the authors that this technology will have much utility as a research tool, and I fully support publication in Nature Communications. I only have a few very minor comments, which the authors and the editors can largely choose to ignore without any substantive drawbacks.

1. In the Abstract, the authors comment on "temporal and spatial" control, and I first believed that the ribozyme was responsible for both. However, the ribozyme is only permits temporal control, whereas spatial expression presumably is due to the promotor. Perhaps the authors could be clearer about this issue.
2. Line 62, the phrase "only few" would be better written as "only a few".
3. Throughout the manuscript, "type 3" should be written as "type III". Likewise, the stems should be named I, II and III.
4. Line 135, I think it is not correct to state that "the integrated aptamer binds its ligand with sub-nanomolar affinity". The published binding data appear to be for the aptamer alone. Once integrated with the hammerhead, I suspect the affinity becomes much poorer.
5. Line 137, the phrase "and well" should be written as "and is well".
6. The term "communication module" should be defined on first use.
7. Line 212, "trashing" should be "thrashing".

Response to points 1. – 7.: Thank you. We agree with the points mentioned and corrected the text passages accordingly.

8. On line 230, I agree that simplicity is the major advantage for worms, but the lack of an adaptive immune response would be far more important for organisms such as humans. Perhaps the authors might add a sentence to note this important problem of protein-based gene regulation systems.

Response: We agree with the reviewer that lack of an immune response is a great advantage of ribozyme-based approaches in humans. We added a sentence to the DISCUSSION section.

9. My greatest concern about the current manuscript is that it took me a lot of time to determine that the authors had not designed the tetracycline-dependent ribozymes, but rather they used sequences that are identical to those published in reference 8. The authors do cite this previous work in the appropriate places, but I think they should add a sentence or two that makes it very clear where these sequences were first published.

Response: As suggested by the reviewer, we have further clarified that the used ribozyme constructs have been introduced by the Suess group for conditional gene expression in human cell lines.

10. Related to Point #9, the authors should have a better graphic for the constructs they used. The RNA constructs are the molecules that other researchers will want to use, but they will have a hard time reconstructing the proper sequence from the fragments presented in Supplementary Figure 1C, or even from the sequences presented in Supplementary Note 1. For example, it is not readily evident from Supplementary Figure 1C what the orientation of the communication modules should be. With a better graphic, readers who wish to use the RNA switches will be much happier.

Response: We added a new scheme to Supplementary Note 1 to make the construct design more clear. We also provide full sequences (5'-3') of the used ribozymes/aptazymes and color-coded the different parts of the aptazyme in Supplementary Note 1. We also improved Supplementary Fig. 2a (former Supplementary Fig. 1c).

11. Many of the references have inaccurate formatting of journal names.

Response: We corrected the formatting of journal names.

Reviewer #2 (Remarks to the Author):

In this manuscript Wurmthaler and colleagues, report the development of an RNA-based genetic system for conditional gene expression in *C. elegans*. This system is a ribozyme switch capable of conferring tetracycline-dependent gene expression upon insertion into the 3'-UTR of the gene of interest. Specifically, the tetracycline-dependent ribozyme switch is constructed by introducing a tetracycline-dependent aptazyme sequence (i.e. tetracycline-binding aptamer sequences) into the 3'-UTR of the gene of interest. Importantly, gene expression can be regulated by tetracycline below its toxicity causing concentration. The system can be used to control tissue specific expression as well as for the study of disease models. The authors tested the efficiency of their system by using the *myo-2::mCherry* reporter and inserted the best performing aptazyme into the 3'-UTR of a ubiquitously expressed mCherry reporter (*icd-1::mCherry*). Transgenic animals expressing the reporter showed an increase mCherry fluorescence following tetracycline treatment. Furthermore, by using tissue-specific promoters, they found that their ribozyme-based system allows tissue specific induction of gene expression. Moreover, the aptazyme-inducible system was used to generate a conditional aggressive nematode model of Huntington's disease by expressing an aggressive Htt109Q pathogenic variant fused to mCherry either ubiquitously (*icd-1* promoter) or specifically in neurons (*rab-3* promoter). Overall, this approach expands the genetic toolbox in *C. elegans*. However, the method is not novel since it is already established and used previously in other systems (Beilstein et al., 2015 ACS synthetic biology; Felletti & Hartig, 2017 RNA).

Comments

Ribozyme-based switches enabling the conditional modulation of gene expression are rarely reported in *C. elegans*. Interestingly, an aptazyme as an artificial riboswitch in the nematode has been described in the PhD thesis of Benedikt Klauser (RNA Synthetic Biology using the Hammerhead Ribozyme: Engineering of Artificial Genetic Switches, 2015). Nevertheless, the utilization of such systems has been extensively reported in bacteria, yeast and mammalian cells, therefore compromising the conceptual novelty of the strategy presented in this study (Auslander et al, Mol

Biosystems, 2010; Wieland et al, Methods, 2012; Felletti et al, 2016, NatCommun, 10.1038/ncomms12834, e.t.c.)

Response: *We agree that ligand-dependent ribozymes have been already developed and utilized for conditional gene expression in unicellular organisms like bacteria, yeast and mammalian cells. However, ribozyme-based switches for conditional gene expression in C. elegans have never been reported. The mentioned PhD thesis of Benedikt Klauser, who actually did his PhD in one of our groups (Prof. Hartig) reports on the activity of a non-controllable hammerhead ribozyme in C. elegans. The unpublished experiments were carried out in 2014 and are the starting point for the presented manuscript. This proof-of-concept experiment showed that ribozymes could in principle be used in nematodes as genetic switches. However, the real challenge within this project was the realization of a ligand-dependent ribozyme. Based on this initial finding, we started to test multiple ligand-dependent ribozyme switches in C. elegans and finally developed the presented tetracycline-dependent ribozyme switch that works efficiently in worms to control gene expression. Considering that one of the greatest limitations of the C. elegans field has been the paucity of conditional systems, we think that our aptazyme-based approach represents a major technical advance for the entire field. In addition to provide a new research tool for C. elegans, our study highlights the utility of ribozyme-based approaches in multicellular organisms, and others will likely adopt the system beyond C. elegans.*

Besides its implementation for ubiquitous and tissue-specific gene expression, the tetracycline-dependent ribozyme switch was validated by generating a nematode model of Huntington's disease. A point of consideration is the lack of statistically significant difference between the number of paralyzed Htt109Q- expressing animals treated with tetracycline and the non-treated controls (Figure 4g). Please, include a statistical analysis. Statistics are missing from the following figures: Figure 4g, Supplementary Figure 1b and 3b-d.

Response: *As suggested by the reviewer, we now include statistics for the data presented in Fig. 4g, Supplementary Fig. 1b, and former Supplementary Fig. 3b-d (now Supplementary Fig. 6b-d).*

To strengthen their claims regarding the robustness of this aptazyme-based genetic tool, the authors should test whether it can efficiently control the temporal expression of genes known to be required for proper embryonic function or neuronal development. In other words, they should test the robustness of their system in a physiological context.

Response: *As suggested by the reviewer, we tested the robustness of our aptazyme-based system in a physiological context. We included a new experiment addressing whether the ribozyme switch allows rescue control of mutant phenotypes in C. elegans. Therefore we used the unc-119(ed3) mutant strain which shows a severe paralysis phenotype due to a developmental defect of the nervous system. Transgenic worms in the unc-119(ed3) mutant background were constructed carrying extrachromosomal arrays with an aptazyme-regulated unc-119 rescue plasmid. Tetracycline treatment significantly rescued the unc-119 mutant phenotype, while no effect was observed in the parental unc-119(ed3) strain. Thus, the aptazyme system is suitable to carry out complementation experiments of mutant alleles in a controlled manner, enabling a precise analysis of gene function during C. elegans development.*

The data are presented in a new Fig. 5, and a short paragraph was added to the RESULTS section. The METHODS section was updated and the new strain information was added to Supplementary Note 2.

The authors should broaden their analyses regarding the side effects of tetracycline in various contexts including ageing, stress conditions, behavior etc. Tetracycline is a strong antibiotic and given the importance of intestinal microbes on health and physiology, it is necessary to validate its effects on worm' physiology.

Response: *We agree with the reviewer and extended our analyses of potential side-effects of tetracycline on worm physiology including life span as well as heat stress adaptation. At the effective concentration used to induce gene expression (10 μ M) tetracycline did not decrease life span or survival of heat-stressed worms. Together with the development and brood size experiments, these data rule out considerable effects on worm physiology by low tetracycline concentrations.*

The new data are provided in Supplementary Fig. 1c, d. A short sentence was added to the RESULTS section. Two paragraphs describing the analyses were added to the METHODS section.

The authors show that HHR (hammerhead ribozyme) insertion into 3' UTR of *myo-2::mcherry* is sufficient for mcherry suppression. However, all constructs/strains used by the authors integrate HHR only into the 3' UTR of *unc-54*. To further evaluate the efficiency of their system, the authors should use translational reporters inserting the HHR into the 3' UTR of genes of interest. Towards this direction, it would be interesting to evaluate the efficiency of their system by targeting GFP instead of mcherry.

Response: *In addition to the *unc-54* 3'UTR we demonstrated functionality also in the *icd-1* 3'UTR. As suggested by the reviewer, we now also show functionality in a translational reporter in the above mentioned *unc-119(ed3)* rescue experiments, in which we inserted the aptazyme in an *unc-119p::Cbr-unc-119::unc-119-3'UTR* plasmid.*

We also constructed a transgenic strain with an aptazyme-regulated GFP reporter (see Figure below). Because we think these data do not add much to our manuscript, we suggest leaving them out. However, if requested by the reviewer or editor, we can add them to the Supplementary section.

Figure 1: *Confocal microscope images of animals carrying *rab-3p::HttQ25::GFP* treated or not with 10 μ M tetracycline from hatch (3 days at 20°C). Neurons of the ventral nerve cord are shown.*

In fig 1d the authors compared transgenic animals injected with myo-2::mcherry active and inactive ribozyme variants. However, the authors do not provide evidence regarding the mRNA levels of mcherry between the two transgenic animals carrying either active or inactive ribozyme variants. Even integrated extrachromosomal arrays of two variants may result in different mRNA levels due to random integration into the genome. Therefore, the authors should perform RT PCR showing the mRNA levels of mcherry between these transgenic animals prior to comparing fluorescence levels of mcherry.

Response: *As suggested by the reviewer, we compared the mCherry mRNA levels in animals carrying active and inactive ribozyme variants in the myo-2p::mCherry reporter. RT-qPCR analysis showed strongly reduced mCherry mRNA levels in worms carrying the active HHR-mCherry construct. These data are consistent with our mechanistic model (Fig. 1a) that self-cleavage of the HHR leads to mRNA decay and thus decreased protein expression.*

The new data are provided in Fig. 1e. A short sentence was added to the RESULTS section. A paragraph describing the RT-qPCR method was added to the METHODS section. Primer sequences were added to Supplementary Note 3.

In fig 2b the authors show tetracycline-induced expression of mcherry in transgenic animals injected only with the active ribozyme variants. To further verify their observations, monitoring tetracycline-induced expression of mcherry in both active and inactive ribozyme variants is required. Towards this direction, transgenic animals expressing only myo-2::mcherry should be used as controls in the absence and presence of tetracycline to exclude any intervention of tetracycline on myo-2 transcriptional levels per se. Within this scope, mRNA levels of myo-2 upon tetracycline supplementation should be monitored. In fig 3a,b authors show tetracycline-induced mcherry expression in neurons, pharynx and muscles. Again, the effect of tetracycline supplementation in both active and inactive ribozyme variants is missing. Furthermore, authors should check whether tetracycline supplementation affects or not the transcript levels of the promoters myo-3 and rab-3 tested.

Response: *As suggested by the reviewer, we included additional control experiments addressing whether the induced mCherry expression is indeed due to the modulation/inhibition of the catalytic activity of the aptazyme by tetracycline. Therefore a new control strain carrying the icd-1p::mCherry reporter with a catalytically inactive aptazyme in the 3'-UTR was constructed. This strain exhibited steady, tetracycline-independent mCherry expression, showing that the inducible mechanism relies on a tetracycline-induced switch based on the activity of the RNA catalyst.*

The new data are provided in Supplementary Fig. 4a, b. A short paragraph was added to the RESULTS section. Strain information was added to Supplementary Note 2.

Moreover, as suggested by the reviewer, we performed additional control experiments to exclude any effects of tetracycline on transcriptional activity of the utilized promoters. For this purpose N2 worms were treated for 3 days with 10 μ M tetracycline and endogenous transcript levels of myo-2, myo-3, rab-3, and icd-1 were analyzed by RT-qPCR. None of the four genes were regulated by tetracycline, indicating that tetracycline induces protein expression only by acting on the inserted aptazyme in the reporter genes.

The new data are provided in Supplementary Fig. 5. A short sentence was added to the RESULTS section. Primer sequences were added to Supplementary Note 3.

Tetracycline causes toxicity in *C. elegans* animals but authors use concentrations that are below the compound toxicity threshold. On the other hand, utilization of this system chronically (i.e. for lifespan experiments) may cause high toxicity due to accumulation of cellular damage. Have authors tested whether this system affects the lifespan and healthspan of wild type animals?

Response: *Please see response to the same issue raised above.*

The authors report that induction of expression is initiated 24h post tetracycline treatment, reaching a peak 48h post treatment. This has three main drawbacks: 1. Expression is not “on” and “off” but is intermediate at least for 48hours until it reaches it’s a peak. This means that the conditions of expression are not well controlled and the researcher is not able to know the exact expression level of the gene of interest. The authors have not tested whether this 48hour-period needed to get the peak in protein expression is dependent on the genetic background and the protein/mRNA of interest and 3. 48h is a long period for a drug-inducible system. It would be interesting if by initially providing much higher drug concentrations, they could rapidly induce gene expression and then by using much milder ones simply maintain expression.

Response: *Our aptazyme approach to control protein expression is based on a post-transcriptional regulation mechanism that is part of the mRNA. It only regulates mRNA stability, but does not regulate other mechanisms that influence protein expression levels like transcriptional activity of promoters or post-translational regulation of protein abundances (protein stability/half life). As many protein abundances are regulated during animal development, we think that the observed peak in expression of an aptazyme-regulated protein will vary from gene to gene and needs to be determined empirically for the protein of interest.*

*Following the reviewers comment concerning the response time of our aptazyme system, we performed additional time- and dose-dependent analyses of induced mRNA levels (on which our system directly acts) using the *icd-1p::mCherry* reporter. RT-qPCR analyses confirmed a rapid tetracycline-induced response on the mRNA level, showing significantly up-regulated mCherry transcript levels (2.2-fold \pm 0.54) already after a 12 hour treatment with 10 μ M tetracycline. The *icd-1p*-driven transcript levels further increased (3.75-fold \pm 0.85) after 48 hours, which is expected considering that *icd-1* is ubiquitously expressed across all life stages of the worm (modENCODE.org).*

The new data are provided in Supplementary 7. A short paragraph was added to the RESULTS section.

The expression of the co-transformation marker is dramatically altered in Figure 1d. This is not expected and may be evidence for multiple secondary effects of this system.

Response: *We are convinced that the unequal distribution of the GFP fluorescence was not caused by tetracycline-induced secondary effects. RT-qPCR analysis showed no effect of tetracycline on GFP expression (see Figure below). A possible explanation could be that transgenic worms carrying extrachromosomal arrays often show a mosaic pattern of expression due to incomplete inheritance of the array (Stinchcomb et al., 1985; Molecular and Cellular Biology, 5 3484-3496). We regret for the misleading figure and now provide better representative microscope images in Fig. 1d.*

Figure 2: GFP expression is not affected by tetracycline treatment in worms carrying *dpy-30p::GFP* and aptazyme-regulated *myo-2p::mCherry*. Diagram shows the fold-induction of GFP mRNA in animals treated with indicated tetracycline concentrations for 12 h, 24 h, and 48 h.

Minor points

Please, provide better quality images to show the body wall muscle- and neuron- specific expression of 3'-UTR aptazyme-regulated mCherry (Figure 3).

Response: We apologize if the quality of figure 3 was not as expected in the presented document. However we suspect that the image quality suffered during conversion of the manuscript since the mentioned images of tissue-selective expression of mCherry in body wall muscles and neurons are of high quality in our files. We will check this issue again in the uploaded file.

In fig 4c-f control conditions are missing.

Response: We have discussed this issue and are convinced that all necessary control conditions in Fig. 4c-f are included. In case that this issue persists, we kindly request to identify specifically the control condition in question.

In Figure 2, panels b and c do not perfectly match to each other. For example, in panel b mCherry is more than doubled in 5μM compared to 1μM treatment, but in panel c this is not evident.

Response: We agree with the reviewer, and repeated the immunoblot analysis. An improved version is now shown in Fig. 2c.

Figure 4: Why did the authors use different 3'UTRs for *icd-1* and *rab-3* promoter?

Response: We designed the ubiquitous expression reporter based on the *icd-1* gene that is ubiquitously expressed. We included the promoter as well as the 3'UTR in order to have a construct with regulatory sequences as natural as possible. The *rab-3* construct is based on plasmid pGH8 (Frokjaer-Jensen et al., 2008; Nature Genetics, 40 1375-1383), which contains an *unc-54* 3'UTR.

Figure 4d and f: Higher magnification images are required for neuronal expression of htt109Q and Htt25Q, to better visualize diffuse expression/aggregates.

Response: *The requested higher magnification images are now provided in the Supplementary Figure 9.*

Figure 4g: Mobility defects must be tested also in the lines with the neuronal expression of Htt109Q and Htt25Q.

Response: *The used thrashing assay is a well-established method to analyze motility defects in C. elegans. This assay is in particular useful for measuring motility defects caused by the dysfunction of the nervous system (Brignull et al., 2006; Journal of Neuroscience, **26** 7597-7606). Hence mobility effects of Htt109Q and Htt25Q are already tested utilizing the thrashing assay presented in the manuscript.*

Please provide information and reference on previous Huntington's Disease model strains in C. elegans (constructs and tissue of expression, as well as motility defects in each case).

Response: *We added a sentence and three more references to the RESULTS section describing previous constitutive polyQ models in C. elegans.*

Please, use proper C. elegans nomenclature.

Response: *We improved the nomenclature throughout the manuscript.*

Reviewers' Comments:

Reviewer #3:

Remarks to the Author:

This is a revised submission of a paper on "Ribozyme-based gene expression switches in *C. elegans*."

Reviewer #2 had many comments, all of which were reasonable.

However, before going over these, the general concern was that the use of ribozymes has been previously shown for bacteria, yeast and mammalian cells. Therefore, it is a fair question in particular for Nature Communication whether it is sufficient to just show that a prior method also works for *C. elegans*. This would not be the case had the ribozyme technology been employed to make a novel discovery in *C. elegans* or that the methodology been fully optimized such that it can be immediately deployed (ie., the widespread use of CrispR or RNAi). For example, a technology for 2019 should not depend on multi-copy expression but rather be more precise CrispR insertion into the target gene of interest.

The authors have a valid point that the *C. elegans* field does not have many conditional regulatory systems so efforts to fully develop the ribozyme system would be much appreciated. The current paper, however points out that the ribozyme system has an unclear dynamic range and was not fully tested so it remains whether this system is limited by tissues and or target gene effects. As a minor point, the authors offer the many advantages of the ribozyme system by comparison to heat shock which despite its limitations is ubiquitous and has a 104-fold dynamic range in inducible gene expression.

Response to Reviewers' comments:

Reviewer #3 (Remarks to the Author):

This is a revised submission of a paper on “Ribozyme-based gene expression switches in *C. elegans*.”

Reviewer #2 had many comments, all of which were reasonable.

However, before going over these, the general concern was that the use of ribozymes has been previously shown for bacteria, yeast and mammalian cells. Therefore, it is a fair question in particular for Nature Communication whether it is sufficient to just show that a prior method also works for *C. elegans*. This would not be the case had the ribozyme technology been employed to make a novel discovery in *C. elegans* or that the methodology been fully optimized such that it can be immediately deployed (ie., the widespread use of CrispR or RNAi). For example, a technology for 2019 should not depend on multi-copy expression but rather be more precise CrispR insertion into the target gene of interest. The authors have a valid point that the *C. elegans* field does not have many conditional regulatory systems so efforts to fully develop the ribozyme system would be much appreciated. The current paper, however points out that the ribozyme system has an unclear dynamic range and was not fully tested so it remains whether this system is limited by tissues and or target gene effects. As a minor point, the authors offer the many advantages of the ribozyme system by comparison to heat shock which despite its limitations is ubiquitous and has a 104-fold dynamic range in inducible gene expression.

Response to Reviewer 3:

As we have detailed before, it takes considerable effort to adopt ribozyme-based expression systems to novel, in particular, multicellular organisms. Within the present work we have implemented ligand-dependent ribozymes in the important research model organism *C. elegans*, for which convenient conditional expression systems are unavailable. In addition, we have also employed this system in order to conditionally express an aggregation-prone and toxic protein, polyQ-Huntingtin, associated with Huntington's disease in humans. It enabled us to show that the aggregating Huntingtin is causing neurotoxicity and we will continue studying protein aggregation and the respective responses in *C. elegans* utilizing the introduced inducible systems. Importantly, this is the first ligand-inducible neurodegenerative Huntington's disease model established in *C. elegans* recapitulating characteristic hallmarks of the disease including inclusion body formation and neurotoxicity. This shows the robustness of our ribozyme-based approach for such applications and, thus, most likely other labs will employ the reported ribozyme approach to generate new inducible *C. elegans* disease models that underlie expression of highly toxic proteins. We have to mention that in this respect multi-copy gene expression is the method of choice since expression by single transgenes is in most cases too weak to reach the toxic threshold needed to generate a robust disease model. We agree with reviewer 3 that a CRISPR insertion into endogenous genes would be interesting and could

be an excellent future application of our switches, but this clearly beyond the scope of our study.

The dynamic range of our switch is not unclear as stated by the reviewer. We have accurately measured this parameter on the protein as well as mRNA level using different techniques including quantitative PCR, immunoblot, fluorescence microscopy and FACS analyses. The results presented in Figure 2, S6, S7 and S8 clearly demonstrate the dynamic range of our expression system. In addition, reviewer 3 mentions that it was not shown that the system works well in all tissues. However, we have convincingly shown that our inducible ligand-dependent ribozymes can be applied in a highly tissue-specific manner. We tested three different tissues (pharynx, body wall muscles, and neurons) and the results presented in Figure 3, 4d, f and S9 c, d clearly show that the system works well in all tested tissues. Moreover, we tested our expression system using a ubiquitous promoter which showed that expression can be turned in most if not all tissues (Figure 2b, S3). The only exception is the germline where no expression was observed. However, this is likely due to the general chromatin silencing of repetitive transgenes as previously observed (Bacaj and Shaham, *Genetics*, 2007) and is thus an aptazyme system-independent effect. In sum, our data indicate that the aptazyme approach functions ubiquitously and is not limited by tissues. We clarify now this issue in the RESULTS section (page 6) and in the legend of Supplementary Fig. 3.

In addition, reviewer 3 mentions the heat shock-responsive control of gene expression as a superior system. However, by applying a heat shock for inducing gene expression, a significant phenotype is triggered that prevents studying gene function in non-stressed conditions. Moreover, a severe drawback of utilizing heat shock-inducible promoters is that it lacks the possibility to achieve tissue-selective expression in a wildtype background. In this respect, we would like to point out again that a significant advantage of our system is the possibility to utilize natural, non-engineered promoters since the switches are simply inserted into the 3'-UTR of the mRNA of interest, preserving the tissue-selective expression of the gene of interest. We added a two sentences to the DISCUSSION section to clarify this issue (page 10).